# Evaluating representations by the complexity of learning low-loss predictors

## Abstract

We consider the problem of evaluating representations of data for use in solving a downstream task. We propose to measure the quality of a representation by the complexity of learning a predictor on top of the representation that achieves low loss on a task of interest. To this end, we introduce two measures: surplus description length (SDL) and $\varepsilon$ sample complexity ($\varepsilon$SC). To compare our methods to prior work, we also present a framework based on plotting the validation loss versus dataset size (the "loss-data" curve). Existing measures, such as mutual information and minimum description length, correspond to slices and integrals along the data-axis of the loss-data curve, while ours correspond to slices and integrals along the loss-axis. This analysis shows that prior methods measure properties of an evaluation dataset of a specified size, whereas our methods measure properties of a predictor with a specified loss. We conclude with experiments on real data to compare the behavior of these methods over datasets of varying size.

## 1 Introduction

One of the first steps in building a machine learning system is selecting a representation of data. Whereas classical machine learning pipelines often begin with feature engineering, the advent of deep learning has led many to argue for pure end-to-end learning where the deep network constructs the features (LeCun et al., 2015). However, huge strides in unsupervised learning (Hénaff et al., 2019; Chen et al., 2020; He et al., 2019; van den Oord et al., 2018; Bachman et al., 2019; Devlin et al., 2019; Liu et al., 2019; Raffel et al., 2019; Brown et al., 2020) have led to a reversal of this trend in the past two years, with common wisdom now recommending that the design of most systems start from a pretrained representation. With this boom in representation learning techniques, practitioners and representation researchers alike have the question: Which representation is best for my task?

This question exists as the middle step of the representation learning pipeline. The first step is representation learning, which consists of training a representation function on a training set using an objective which may be supervised or unsupervised. The second step, which this paper considers, is representation evaluation. In this step, one uses a measure of representation quality and a labeled *evaluation dataset* to see how well the representation performs. The final step is deployment, in which the practitioner or researcher puts the learned representation to use. Deployment could involve using the representation on a stream of user-provided data to solve a variety of end tasks (LeCun, 2015), or simply releasing the trained weights of the representation function for general use. In the same way that BERT (Devlin et al., 2019) representations have been applied to a whole host of problems, the task or amount of data available in deployment might differ from the evaluation phase.

We take the position that the best representation is the one which allows for the most *efficient* learning of a predictor to solve the task. We will measure efficiency in terms of either number of samples or information about the optimal predictor contained in the samples. This position is motivated by practical concerns; the more labels that are needed to solve a task in the deployment phase, the more expensive to use and the less widely applicable a representation will be.

We build on a substantial and growing body of literature that attempts to answer the question of which representation is best. Simple, traditional means of evaluating representations, such as the validation accuracy of linear probes (Ettinger et al., 2016; Shi et al., 2016; Alain & Bengio, 2016), have been widely criticized (Hénaff et al., 2019; Resnick et al., 2019). Instead, researchers have taken

up a variety of alternatives such as the validation accuracy (VA) of nonlinear probes (Conneau et al., 2018; Hénaff et al., 2019), mutual information (MI) between representations and labels (Bachman et al., 2019; Pimentel et al., 2020), and minimum description length (MDL) of the labels conditioned on the representations (Blier & Ollivier, 2018; Yogatama et al., 2019; Voita & Titov, 2020).

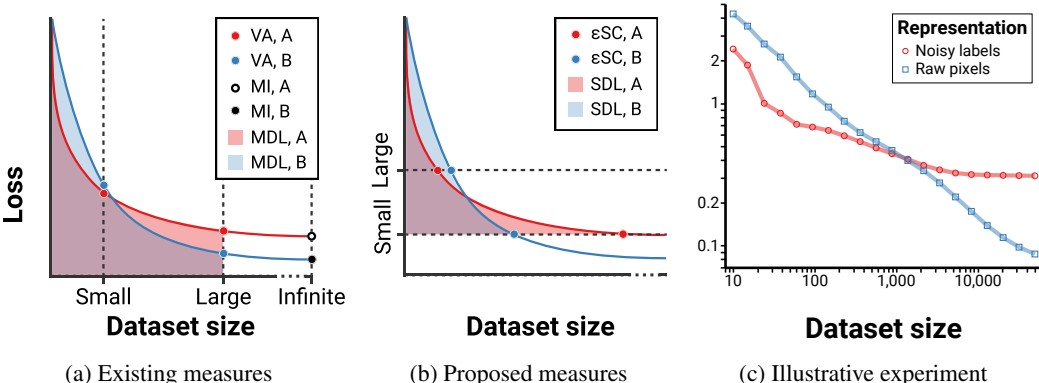

(a) Existing measures     (b) Proposed measures     (c) Illustrative experiment

Figure 1: Each measure for evaluating representation quality is a simple function of the "loss-data" curve shown here, which plots validation loss of a probe against evaluation dataset size. **Left:** Validation accuracy (VA), mutual information (MI), and minimum description length (MDL) measure properties of a given dataset, with VA measuring the loss at a finite amount of data, MI measuring it at infinity, and MDL integrating it from zero to $n$. This dependence on dataset size can lead to misleading conclusions as the amount of available data changes. **Middle:** Our proposed methods instead measure the complexity of learning a predictor with a particular loss tolerance. $\varepsilon$ sample complexity ($\varepsilon$SC) measures the number of samples required to reach that loss tolerance, while surplus description length (SDL) integrates the surplus loss incurred above that tolerance. Neither depends on the dataset size. **Right:** A simple example task which illustrates the issue. One representation, which consists of noisy labels, allows quick learning, while the other supports low loss in the limit of data. Evaluating either representation at a particular dataset size risks drawing the wrong conclusion.

We find that these methods all have clear limitations. As can be seen in Figure 1, VA and MDL are liable to choose different representations for the same task when given evaluation datasets of different sizes. Instead we want an evaluation measure which depends on the data *distribution*, not a particular dataset or dataset size. Furthermore, VA and MDL lack a predefined notion of success in solving a task. In combination with small evaluation datasets, these measures may lead to premature evaluation by producing a judgement even when there is not enough data to solve the task or meaningfully distinguish one representation from another. Meanwhile, MI measures the lowest loss achievable by any predictor irrespective of the complexity of learning it. We note that while these methods do not correspond to our notion of best representation, they may be correct for different notions of "best".

To eliminate these issues, we propose two measures. In both of our measures, the user must specify a tolerance $\varepsilon$ so that a population loss of less than $\varepsilon$ qualifies as solving the task. The first measure is the *surplus description length* (SDL) which modifies the MDL to measure the complexity of learning an $\varepsilon$-loss predictor rather than the complexity of the labels in the evaluation dataset. The second is the $\varepsilon$-*sample complexity* ($\varepsilon$SC) which measures the sample complexity of learning an $\varepsilon$-loss predictor.

To facilitate our analysis, we also propose a framework called the *loss-data framework*, illustrated in Figure 1, that plots the validation loss against the evaluation dataset size (Talmor et al., 2019; Yogatama et al., 2019; Voita & Titov, 2020). This framework simplifies comparisons between measures. Prior work measures integrals (MDL) and slices (VA and MI) along the data-axis. Our work proposes instead measuring integrals (SDL) and slices ($\varepsilon$SC) along the loss-axis. This illustrates how prior work makes tacit choices about the function to learn based on the choice of dataset size. Our work instead makes an explicit, interpretable choice of threshold $\varepsilon$ and measures the complexity of solving the task to $\varepsilon$ error. We experimentally investigate the behavior of these methods, illustrating the sensitivity of VA and MDL, and the robustness of SDL and $\varepsilon$SC, to dataset size.

**Efficient implementation.** To enable reproducible and efficient representation evaluation for representation researchers, we have developed a highly optimized open source Python package (see

supplementary materials). This package enables construction of loss-data curves with arbitrary representations and datasets and is library-agnostic, supporting representations and learning algorithms implemented in any Python ML library. By leveraging the JAX library (Bradbury et al., 2018) to parallelize the training of probes on a single accelerator, our package constructs loss-data curves in around two minutes on one GPU.

## 2  THE LOSS-DATA FRAMEWORK FOR REPRESENTATION EVALUATION

In this section we formally present the representation evaluation problem, define our loss-data framework, and show how prior work fits into the framework.

**Notation.**   We use bold letters to denote random variables. A supervised learning problem is defined by a joint distribution $\mathcal{D}$ over observations and labels $(\mathbf{X}, \mathbf{Y})$ in the sample space $\mathcal{X} \times \mathcal{Y}$ with density denoted by $p$. Let the random variable $\mathbf{D}^n$ be a sample of $n$ i.i.d. $(\mathbf{X}, \mathbf{Y})$ pairs, realized by $D^n = (X^n, Y^n) = \{(x_i, y_i)\}_{i=1}^n$. Let $\mathcal{R}$ denote a representation space and $\phi : \mathcal{X} \to \mathcal{R}$ a representation function. The methods we consider all use parametric probes, which are neural networks $\hat{p}_\theta : \mathcal{R} \to P(\mathcal{Y})$ parameterized by $\theta \in \mathbb{R}^d$ that are trained on $D^n$ to estimate the conditional distribution $p(y \mid x)$. We often abstract away the details of learning the probe by simply referring to an algorithm $\mathcal{A}$ which returns a predictor: $\hat{p} = \mathcal{A}(\phi(D^n))$. Abusing notation, we denote the composition of $\mathcal{A}$ with $\phi$ by $\mathcal{A}_\phi$. Define the population loss and the expected population loss for $\hat{p} = \mathcal{A}_\phi(D^n)$, respectively as

$$L(\mathcal{A}_\phi, D^n) = \mathop{\mathbb{E}}_{(\mathbf{X},\mathbf{Y})} - \log \hat{p}(\mathbf{Y} \mid \mathbf{X}), \qquad L(\mathcal{A}_\phi, n) = \mathop{\mathbb{E}}_{\mathbf{D}^n} L(\mathcal{A}_\phi, \mathbf{D}^n). \qquad (1)$$

In this section we will focus on population quantities, but note that any algorithmic implementation must replace these by their empirical counterparts.

**The representation evaluation problem.**   The representation evaluation problem asks us to define a real-valued measurement of the quality of a representation $\phi$ for solving solving the task defined by $(\mathbf{X}, \mathbf{Y})$. Explicitly, each method defines a real-valued function $m(\phi, \mathcal{D}, \mathcal{A}, \Psi)$ of a representation $\phi$, data distribution $\mathcal{D}$, probing algorithm $\mathcal{A}$, and some method-specific set of hyperparameters $\Psi$. By convention, smaller values of the measure $m$ correspond to better representations. Defining such a measurement allows us to compare different representations.

### 2.1  DEFINING THE LOSS-DATA FRAMEWORK.

The loss-data framework is a lens through which we contrast different measures of representation quality. The key idea, demonstrated in Figure 1, is to plot the loss $L(\mathcal{A}_\phi, n)$ against the dataset size $n$. Explicitly, at each $n$, we train a probing algorithm $\mathcal{A}$ using a representation $\phi$ to produce a predictor $\hat{p}$, and then plot the loss of $\hat{p}$ against $n$. Similar analysis has appeared in Voita & Titov (2020); Yogatama et al. (2019); Talmor et al. (2019). We can represent each of the prior measures as points on the curve at fixed $x$ (VA, MI) or integrals of the curve along the $x$-axis (MDL). Our measures correspond to evaluating points at fixed $y$ ($\varepsilon$SC) and integrals along the $y$-axis (SDL).

### 2.2  EXISTING METHODS IN THE LOSS-DATA FRAMEWORK

**Nonlinear probes with limited data.**   A simple strategy for evaluating representations is to choose a probe architecture and train it on a limited amount of data from the task and representation of interest (Hénaff et al., 2019; Zhang & Bowman, 2018). On the loss-data curve, this corresponds to evaluation at $x = n$, so that

$$m_{\text{VA}}(\phi, \mathcal{D}, \mathcal{A}, n) = L(\mathcal{A}_\phi, n). \qquad (2)$$

**Mutual information.**   Mutual information (MI) between a representation $\phi(\mathbf{X})$ and targets $\mathbf{Y}$ is another often-proposed metric for learning and evaluating representations (Pimentel et al., 2020; Bachman et al., 2019). In terms of entropy, mutual information is equivalent to the information gain about $\mathbf{Y}$ from knowing $\phi(\mathbf{X})$:

$$I(\phi(\mathbf{X}); \mathbf{Y}) = H(\mathbf{Y}) - H(\mathbf{Y} \mid \phi(\mathbf{X})). \qquad (3)$$

In general mutual information is intractable to estimate for high-dimensional or continuous-valued variables (McAllester & Stratos, 2020), and a common approach is to use a very expressive model for $\hat{p}$ and maximize a variational lower bound:

$$I(\phi(\mathbf{X}); \mathbf{Y}) \geq H(\mathbf{Y}) + \mathop{\mathbb{E}}_{(\mathbf{X}, \mathbf{Y})} \log \hat{p}(\mathbf{Y} \mid \phi(\mathbf{X})). \tag{4}$$

Since $H(\mathbf{Y})$ is not a function of the parameters, maximizing the lower bound is equivalent to minimizing the negative log-likelihood. Moreover, if we assume that $\hat{p}$ is expressive enough to represent $p$ and take $n \to \infty$, this inequality becomes tight. As such, MI estimation can be seen a special case of nonlinear probes as described above, where instead of choosing some particular setting of $n$ we push it to infinity. We formally define the mutual information measure of a representation as

$$m_{\mathrm{MI}}(\phi, \mathcal{D}, \mathcal{A}) = \lim_{n \to \infty} L(\mathcal{A}_\phi, n). \tag{5}$$

A decrease in this measure reflects an increase in the mutual information. On the loss-data curve, this corresponds to evaluation at $x = \infty$.

**Minimum description length.** Recent studies (Yogatama et al., 2019; Voita & Titov, 2020) propose using the Minimum Description Length (MDL) principle (Rissanen, 1978; Grünwald, 2004) to evaluate representations. These works use an online or prequential code (Blier & Ollivier, 2018) to encode the labels given the representations. The codelength $\ell$ of $Y^n$ given $\phi(X^n)$ is then defined as

$$\ell(Y^n \mid \phi(X^n)) = -\sum_{i=1}^{n} \log \hat{p}_i(y_i \mid \phi(x_i)), \tag{6}$$

where $\hat{p}_i$ is the output of running a pre-specified algorithm $\mathcal{A}$ on the dataset up to element $i$: $\hat{p}_i = \mathcal{A}_\phi(X_{1:i}^n, Y_{1:i}^n)$. Taking an expectation over the sampled datasets for each $i$, we define a population variant of the MDL measure (Voita & Titov, 2020) as

$$m_{\mathrm{MDL}}(\phi, \mathcal{D}, \mathcal{A}, n) = \mathbb{E}\left[\ell(\mathbf{Y}^n \mid \phi(\mathbf{X}^n))\right] = \sum_{i=1}^{n} L(\mathcal{A}, i). \tag{7}$$

Thus, $m_{\mathrm{MDL}}$ measures the area under the loss-data curve on the interval $x \in [0, n]$.

## 3    LIMITATIONS OF EXISTING METHODS

Each of the prior methods, VA, MDL, and MI, have limitations that we attempt to solve with our methods. In this section we present these limitations.

### 3.1    SENSITIVITY TO DATASET SIZE IN VA AND MDL

As seen in Section 2.2, the representation quality measures of VA and MDL both depend on $n$, the size of the evaluation dataset. Because of this dependence, the ranking of representations given by these evaluation metrics can change as $n$ increases. Choosing to deploy one representation rather than another by comparing these metrics at arbitrary $n$ may lead to premature decisions in the machine learning pipeline since a larger dataset could give a different ordering.

**A theoretical example.** Let $s \in \{0, 1\}^d$ be a fixed binary vector and consider a data generation process where the $\{0, 1\}$ label of a data point is given by the parity on $s$, i.e., $y_i = \langle x_i, s \rangle \bmod 2$ where $y_i \in \{0, 1\}$ and $x_i \in \{0, 1\}^d$. Let $Y^n = \{y_i\}_{i=1}^{n}$ be the given labels and consider the following two representations: (1) Noisy label: $z_i = \langle x_i, s \rangle + e_i \bmod 2$, where $e_i \in \{0, 1\}$ is a random bit with bias $\alpha < 1/2$, and (2) Raw data: $x_i$.

For the noisy label representation, guessing $y_i = z_i$ achieves validation accuracy of $1 - \alpha$ for any $n$, which, is information-theoretically optimal. On the other hand, the raw data representation will achieve perfect validation accuracy once the evaluation dataset contains $d$ linearly independent $x_i$'s. In this case, Gaussian elimination will exactly recover $s$. The probability that a set of $n > d$ random vectors in $\{0, 1\}^d$ does not contain $d$ linearly independent vectors decreases exponentially in $n - d$. Hence, the expected validation accuracy for $n$ sufficiently larger than $d$ will be exponentially close

to 1. As a result, the representation ranking given by validation accuracy and description length favors the noisy label representation when $n \ll d$, but the raw data representation will be much better in these metrics when $n \gg d$. This can be misleading. Although this is a concocted example for illustration purposes, our experiments in Section 5 show dependence of representation rankings on $n$.

## 3.2 INSENSITIVITY TO REPRESENTATION QUALITY & COMPUTATIONAL COMPLEXITY IN MI

MI considers the lowest validation loss achievable with the given representation and ignores any concerns about statistical or computational complexity of achieving such accuracy. This leads to some counterintuitive properties which make MI an undesirable metric:

1. MI is insensitive to statistical complexity. Two random variables which are perfectly predictive of one another have maximal MI, though their relationship may be sufficiently complex that it requires exponentially many samples to verify (McAllester & Stratos, 2020).

2. MI is insensitive to computational complexity. For example, the mutual information between an intercepted encrypted message and the enemy's plan is high (Shannon, 1948; Xu et al., 2020), despite the extreme computational cost required to break the encryption.

3. MI is insensitive to representation. By the data processing inequality (Cover & Thomas, 2006), *any* $\phi$ applied to $\mathbf{X}$ can only decrease its mutual information with $\mathbf{Y}$; no matter the query, MI always reports that the raw data is at least as good as the best representation.

## 3.3 LACK OF A PREDEFINED NOTION OF SUCCESS

All three prior methods lack a predefined notion of successfully solving a task and will always return some ordering of representations. When the evaluation dataset is too small or all of the representations are poor, it may be that no representation can yet solve the task. Since the order of representations can change as more data is added, any judgement would be premature. Indeed, there is often an implicit minimum requirement for the loss a representation should achieve to be considered meaningful. As we show in the next section, our methods makes this requirement explicit.

## 4 SURPLUS DESCRIPTION LENGTH & $\varepsilon$ SAMPLE COMPLEXITY

The methods discussed above measure a property of the data, such as the attainable accuracy on $n$ points, by learning an unspecified function. Instead, we propose to precisely define the function of interest and measure its complexity using data. Fundamentally we shift from making a statement about the inputs of an algorithm, like VA and MDL do, to a statement about the outputs.

### 4.1 SURPLUS DESCRIPTION LENGTH (SDL)

Imagine trying to efficiently encode a large number of samples of a random variable $\mathbf{e}$ which takes values in $\{1 \ldots K\}$ with probability $p(\mathbf{e})$. An optimal code for these events has expected length[1] $\mathbb{E}[\ell(\mathbf{e})] = \mathbb{E}_{\mathbf{e}}[-\log p(\mathbf{e})] = H(\mathbf{e})$. If this data is instead encoded using a probability distribution $\hat{p}$, the expected length becomes $H(\mathbf{e}) + D_{\mathrm{KL}}(p \,||\, \hat{p})$. We call $D_{\mathrm{KL}}(p \,||\, \hat{p})$ the *surplus description length* (SDL) from encoding according to $\hat{p}$ instead of $p$:

$$D_{\mathrm{KL}}(p \,||\, \hat{p}) = \mathop{\mathbb{E}}_{\mathbf{e} \sim p} \left[ \log p(\mathbf{e}) - \log \hat{p}(\mathbf{e}) \right]. \tag{8}$$

When the true distribution $p$ is a delta, the entire length of a code under $\hat{p}$ is surplus since $\log 1 = 0$.

Recall that the prequential code for estimating MDL computes the description length of the labels given observations in a dataset by iteratively creating tighter approximations $\hat{p}_1 \ldots \hat{p}_n$ and integrating the area under the curve. Examining Equation (7), we see that

$$m_{\mathrm{MDL}}(\phi, \mathcal{D}, \mathcal{A}, n) = \sum_{i=1}^{n} L(\mathcal{A}_\phi, i) \geq \sum_{i=1}^{n} H(\mathbf{Y} \mid \phi(\mathbf{X})). \tag{9}$$

---

[1]in nats

If $H(\mathbf{Y} \mid \phi(\mathbf{X})) > 0$, MDL grows without bound as the size of the evaluation dataset $n$ increases.

Instead, we propose to measure the complexity of a learned predictor $p(\mathbf{Y} \mid \phi(\mathbf{X}))$ by computing the surplus description length of encoding an infinite stream of data according to the online code instead of the true conditional distribution.

**Definition 1** (Surplus description length of online codes). *Given random variables* $\mathbf{X}, \mathbf{Y} \sim \mathcal{D}$, *a representation function* $\phi$, *and a learning algorithm* $\mathcal{A}$, *define*

$$m_{\mathrm{SDL}}(\phi, \mathcal{D}, \mathcal{A}) = \sum_{i=1}^{\infty} \Big[ L(\mathcal{A}_\phi, i) - H(\mathbf{Y} \mid \mathbf{X}) \Big]. \tag{10}$$

We generalize this definition to measure the complexity of learning an approximating conditional distribution with loss $\varepsilon$, rather than the true conditional distribution only:

**Definition 2** (Surplus description length of online codes with an arbitrary baseline). *Take random variables* $\mathbf{X}, \mathbf{Y} \sim \mathcal{D}$, *a representation function* $\phi$, *a learning algorithm* $\mathcal{A}$, *and a loss tolerance* $\varepsilon \geq H(\mathbf{Y} \mid \mathbf{X})$. *Let* $[c]_+$ *denote* $\max(0, c)$ *and then we define*

$$m_{\mathrm{SDL}}(\phi, \mathcal{D}, \mathcal{A}, \varepsilon) = \sum_{i=1}^{\infty} \Big[ L(\mathcal{A}_\phi, i) - \varepsilon \Big]_+. \tag{11}$$

In our framework, the surplus description length corresponds to computing the area between the loss-data curve and a baseline set by $y = \varepsilon$. Whereas MDL measures the complexity of a sample of $n$ points, SDL measures the complexity of a function which solves the task to $\varepsilon$ tolerance.

**Estimating the SDL.**    Naively computing SDL would require unbounded data and the estimation of $L(\mathcal{A}_\phi, i)$ for every $i$. However, if we assume that algorithms are monotonically improving so that $L(\mathcal{A}, i + 1) \leq L(\mathcal{A}, i)$, SDL only depends on $i$ up to the first point where $L(\mathcal{A}, n) \leq \varepsilon$. Approximating this integral can be done efficiently by taking a log-uniform partition of the dataset size and computing the Riemann sum as in Voita & Titov (2020). Crucially, if the tolerance $\varepsilon$ is set too low or the maximum amount of available data is insufficient, an implementation is able to report that the given complexity estimate is only a lower bound. In Appendix A we provide a detailed algorithm for estimating SDL, along with a theorem proving its data requirements.

### 4.2   $\varepsilon$ SAMPLE COMPLEXITY ($\varepsilon$SC)

In addition to surplus description length we introduce a second, conceptually simpler measure of representation quality: $\varepsilon$ sample complexity.

**Definition 3** (Sample complexity of an $\varepsilon$-loss predictor). *Given random variables* $\mathbf{X}, \mathbf{Y} \sim \mathcal{D}$, *a representation function* $\phi$, *a learning algorithm* $\mathcal{A}$, *and a loss tolerance* $\varepsilon \geq H(\mathbf{Y} \mid \phi(\mathbf{X}))$, *define*

$$m_{\varepsilon\mathrm{SC}}(\phi, \mathcal{D}, \mathcal{A}, \varepsilon) = \min \Big\{ n \in \mathbb{N} : L(\mathcal{A}_\phi, n) \leq \varepsilon \Big\}. \tag{12}$$

Sample complexity measures the complexity of learning an $\varepsilon$-loss predictor by the number of samples it takes to find it. In our framework, sample complexity corresponds to taking a horizontal slice of the loss-data curve at $y = \varepsilon$, analogous to VA. VA makes a statement about the data (by setting $n$) and reports the accuracy of some function given that data. In contrast, sample complexity specifies the desired function and determines its complexity by how many samples are needed to learn it.

**Estimating the $\varepsilon$SC.**    Given an assumption that algorithms are monotonically improving such that $L(\mathcal{A}, n + 1) \leq L(\mathcal{A}, n)$, $\varepsilon$SC can be estimated efficiently. With $n$ finite samples in the dataset, an algorithm may estimate $\varepsilon$SC by splitting the data into $k$ uniform-sized bins and estimating $L(\mathcal{A}, ik/n)$ for $i \in \{1 \ldots k\}$. By recursively performing this search on the interval which contains the transition from $L > \varepsilon$ to $L < \varepsilon$, we can rapidly reach a precise estimate or report that $m_{\varepsilon\mathrm{SC}}(\phi, \mathcal{D}, \mathcal{A}, \varepsilon) > n$. A more detailed examination of the algorithmic considerations of estimating $\varepsilon$SC is in Appendix B.

**Using objectives other than negative log-likelihood.**    Our exposition of $\varepsilon$SC uses negative log-likelihood for consistency with other methods, such as MDL, which require it. However, it is straightforward to extend $\varepsilon$SC to work with whatever objective function is desired under the assumption that said objective is monotone with increasing data when using algorithm $\mathcal{A}$.

### 4.3 SETTING $\varepsilon$

A value for the threshold $\varepsilon$ corresponds to the set of $\varepsilon$-loss predictors that a representation should make easy to learn. Choices of $\varepsilon \geq H(\mathbf{Y} \mid \mathbf{X})$ represent attainable functions, while selecting $\varepsilon < H(\mathbf{Y} \mid \mathbf{X})$ leads to unbounded SDL and $\varepsilon$SC for any choice of the algorithm $\mathcal{A}$.

For evaluating representation learning methods in the research community, we recommend using SDL and establishing benchmarks which specify (1) a downstream task, in the form of a dataset; (2) a criterion for success, in the form of a setting of $\varepsilon$; (3) a standard probing algorithm $\mathcal{A}$. The setting of $\varepsilon$ can be done by training a large model on the raw representation of the full dataset and using its validation loss as $\varepsilon$ when evaluating other representations. This guarantees that $\varepsilon \geq H(\mathbf{Y} \mid \mathbf{X})$ and the task is feasible with a good representation; in turn, this ensures that SDL is bounded.

In practical applications, $\varepsilon$ should be a part of the design specification for a system. As an example, a practitioner might know that an object detection system with 80% per-frame accuracy is sufficient and labels are expensive. For this task, the best representation would be one which enables the most sample efficient learning of a predictor with error $\varepsilon = 0.2$ using a $0 - 1$ loss.

| n | Representation | CIFAR | Pixels | VAE |
|---|---|---|---|---|
| 60 | Val loss | 0.88 | 1.54 | **0.70** |
| | MDL | 122.75 | 147.34 | **93.8** |
| | SDL, $\varepsilon$=1 | 65.33 | > 87.34 | **40.75** |
| | SDL, $\varepsilon$=0.2 | > 110.75 | > 135.34 | **> 81.8** |
| | $\varepsilon$SC, $\varepsilon$=1 | 60 | > 60.0 | **38** |
| | $\varepsilon$SC, $\varepsilon$=0.2 | > 60.0 | > 60.0 | > 60.0 |
| 20398 | Val loss | **0.05** | 0.11 | 0.14 |
| | MDL | **1607.95** | 3876.88 | 3360.49 |
| | SDL, $\varepsilon$=1 | 65.33 | 93.06 | **40.75** |
| | SDL, $\varepsilon$=0.2 | **153.49** | 800.16 | 278.95 |
| | $\varepsilon$SC, $\varepsilon$=1 | 60 | 147 | **38** |
| | $\varepsilon$SC, $\varepsilon$=0.2 | **884** | 8322 | 3395 |

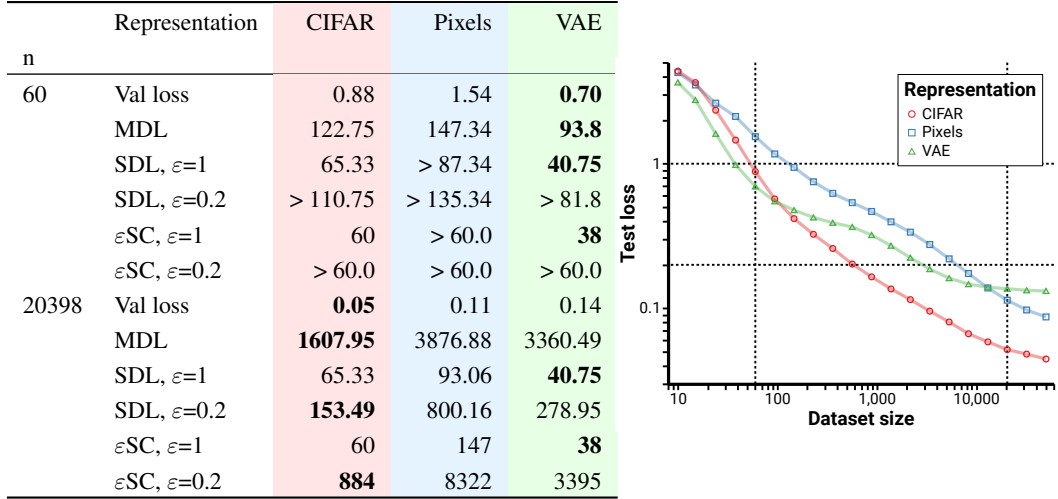

Figure 2: Results using three representations on the MNIST dataset.

## 5 EXPERIMENTS

We empirically show the behavior of VA, MDL, SDL, and $\varepsilon$SC with two sets of experiments on real data. For the first, shown in Figure 2, we evaluate three representations on MNIST classification: (1) the last hidden layer of a small convolutional network pretrained on CIFAR-10; (2) raw pixels; and (3) a variational autoencoder (VAE) (Kingma & Welling, 2014; Rezende et al., 2014) trained on MNIST. For the second experiment, shown in Figure 3, we compare the representations given by different layers of a pretrained ELMo model (Peters et al., 2018) using the part-of-speech task introduced by Hewitt & Liang (2019) and implemented by Voita & Titov (2020) with the same probe architecture and other hyperparameters as those works. Note that in each experiment we omit MI as for any finite amount of data, the MI measure is the same as validation loss. Details of the experiments, including representation training, probe architectures, and hyperparameters, are available in Appendix C.

These experiments demonstrate that the issue of sensitivity to evaluation dataset size in fact occurs in practice, both on small problems (Figure 2) and at scale (Figure 3): VA and MDL both choose different representations when given evaluation sets of different sizes. Because these measures are a function of the dataset size, making a decision about which representation to use with a small evaluation dataset would be premature. By contrast, SDL and $\varepsilon$SC are functions only of the data *distribution*, not a finite sample. Once they measure the complexity of learning an $\varepsilon$-loss function, that measure is invariant to the size of the evaluation dataset. Crucially, since these measures contain

a notion of success in solving a task, they are able to avoid the issue of premature evaluation and notify the user if there is insufficient data to evaluate and return a lower bound instead.

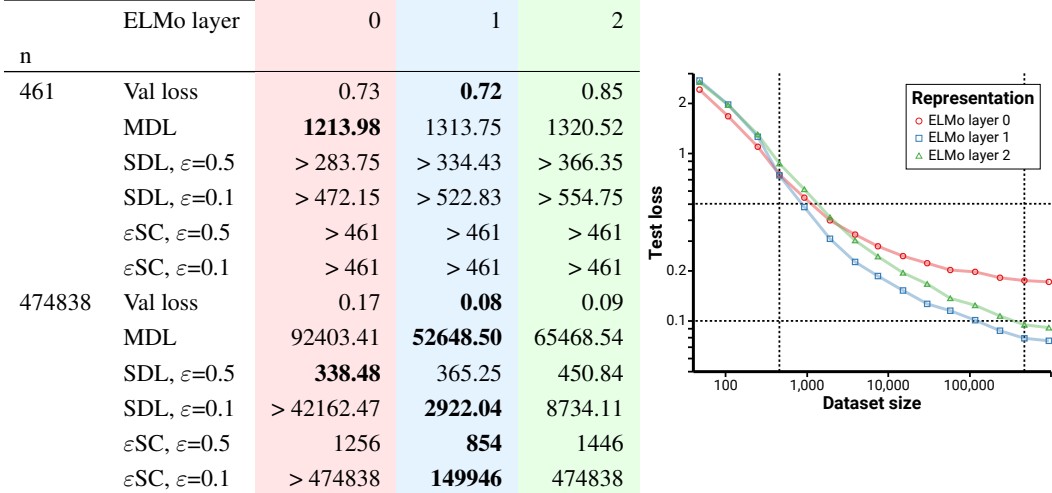

| n | ELMo layer | 0 | 1 | 2 |
|---|---|---|---|---|
| 461 | Val loss | 0.73 | **0.72** | 0.85 |
| | MDL | **1213.98** | 1313.75 | 1320.52 |
| | SDL, $\varepsilon$=0.5 | > 283.75 | > 334.43 | > 366.35 |
| | SDL, $\varepsilon$=0.1 | > 472.15 | > 522.83 | > 554.75 |
| | $\varepsilon$SC, $\varepsilon$=0.5 | > 461 | > 461 | > 461 |
| | $\varepsilon$SC, $\varepsilon$=0.1 | > 461 | > 461 | > 461 |
| 474838 | Val loss | 0.17 | **0.08** | 0.09 |
| | MDL | 92403.41 | **52648.50** | 65468.54 |
| | SDL, $\varepsilon$=0.5 | **338.48** | 365.25 | 450.84 |
| | SDL, $\varepsilon$=0.1 | > 42162.47 | **2922.04** | 8734.11 |
| | $\varepsilon$SC, $\varepsilon$=0.5 | 1256 | **854** | 1446 |
| | $\varepsilon$SC, $\varepsilon$=0.1 | > 474838 | **149946** | 474838 |

Figure 3: Results using three representations on a part of speech classification task.

# 6    RELATED WORK

Zhang & Bowman (2018) and Hewitt & Liang (2019) propose random baselines for linguistic tasks to provide context for how much linguistic structure is readily accessible in representations. To show separation between the validation accuracy achieved by these random baselines and representations pretrained on genuine linguistic labels, they have to limit the amount of training data or restrict the capacity of probes. As an alternative, Voita & Titov (2020) propose using the MDL framework, which measures the description length of the labels given the observations, to demonstrate the separation between pretrained representations and random baselines. An earlier work by Yogatama et al. (2019) also uses prequential codes to evaluate representations for linguistic tasks. Foundational work by Blier & Ollivier (2018) introduces prequential codes as a measure of the complexity of a deep learning model. Talmor et al. (2019) look at the loss-data curve (called "learning curve" in their work) and use a weighted average of the validation loss at various training set sizes to evaluate representations.

# 7    DISCUSSION

In this work we have introduced the loss-data framework for comparing representation evaluation measures and used it to diagnose the issue of sensitivity to evaluation dataset size in the validation accuracy and minimum description length measures. We proposed two measures, surplus description length and $\varepsilon$ sample complexity, which eliminate this issue by measuring the complexity of learning a predictor which solves the task of interest to $\varepsilon$ tolerance. Empirically we showed that sensitivity to evaluation dataset size occurs in practice for VA and MDL, while SDL and $\varepsilon$SC are robust to the amount of available data and are able to report when it is insufficient to make a judgment.

Each of these measures depends on a choice of algorithm $\mathcal{A}$, including hyperparameters such as probe architecture, which could make the evaluation procedure less robust. To alleviate this, future work might consider a set of algorithms $A = \{\mathcal{A}_i\}_{i=1}^{K}$ and a method of combining them, such as the model switching technique of Blier & Ollivier (2018); Erven et al. (2012) or a Bayesian prior.

Finally, while existing measures such as VA, MI, and MDL do not measure our notion of the best representation for a task, under other settings they may be the correct choice. For example, if only a fixed set of data will ever be available, selecting representations using VA might be a reasonable choice; and if unbounded data is available for free, perhaps MI is the most appropriate measure. However, in many cases the robustness and interpretability offered by SDL and $\varepsilon$SC make them a practical choice for practitioners and representation researchers alike.

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

## APPENDIX A ALGORITHMIC DETAILS FOR ESTIMATING SURPLUS DESCRIPTION LENGTH

Recall that the SDL is defined as

$$m_{\text{SDL}}(\phi, \mathcal{D}, \mathcal{A}, \varepsilon) = \sum_{n=1}^{\infty} \Big[ L(\mathcal{A}_\phi, n) - \varepsilon \Big]_+ \tag{13}$$

For simplicity, we assume that $L$ is bounded in $[0, 1]$. Note that this can be achieved by truncating the cross-entropy loss.

---

**Algorithm 1:** Estimate surplus error

---

**Input:** tolerance $\varepsilon$, max iterations $M$, number of datasets $K$, representation $\phi$, data distribution $\mathcal{D}$, algorithm $\mathcal{A}$

**Output:** Estimate $\hat{m}$ of $m(\phi, \mathcal{D}, \varepsilon, \mathcal{A})$ and indicator $I$ of whether this estimate is tight or lower bound

---

Sample $K$ datasets $D_M^k \sim \mathcal{D}$ of size $M + 1$
**for** $n = 1$ **to** $M$ **do**
    For each $k \in [K]$, run $\mathcal{A}$ on $D_M^k[1:n]$ to produce a predictor $\hat{p}_n^k$
    Take $K$ test samples $(x_k, y_k) = D_M^k[M + 1]$
    Evaluate $\hat{L}_n = \frac{1}{K} \sum_{k=1}^K \ell(\hat{p}_n^k, x_k, y_k)$
Set $\hat{m} = \sum_{n=1}^M [\hat{L}_n - \varepsilon]_+$
**if** $\hat{L}_M \leq \varepsilon/2$ **then** Set $I = \texttt{tight}$ **else** Set $I = \texttt{lower bound}$;
**return** $\hat{m}, I$

---

In our experiments we replace $D_M^k[1:n]$ of Algorithm 1 with sampled subsets of size $n$ from a single evaluation dataset. Additionally, we use between 10 and 20 values of $n$ instead of evaluating $L(\mathcal{A}_\phi, n)$ at every integer between 1 and $M$. This strategy, also used by Blier & Ollivier (2018) and Voita & Titov (2020), corresponds to the description length under a code which updates only periodically during transmission of the data instead of after every single point.

**Theorem 4.** *Let the loss function $L$ be bounded in $[0, 1]$ and assume that it is decreasing in $n$. With $(M + 1)K$ datapoints, if the sample complexity is less than $M$, the above algorithm returns an estimate $\hat{m}$ such that with probability at least $1 - \delta$*

$$|\hat{m} - m(\phi, \mathcal{D}, \varepsilon, \mathcal{A})| \leq M \sqrt{\frac{\log(2M/\delta)}{2K}}. \tag{14}$$

*If $K \geq \frac{\log(1/\delta)}{2\varepsilon^2}$ and the algorithm returns* `tight` *then with probability at least $1 - \delta$ the sample complexity is less than $M$ and the above bound holds.*

*Proof.* First we apply a Hoeffding bound to show that each $\hat{L}_n$ is estimated well. For any $n$, we have

$$P\left( |\hat{L}_n - L(\mathcal{A}_\phi, n)| > \sqrt{\frac{\log(2M/\delta)}{2K}} \right) \leq 2 \exp\left( -2K \frac{\log(2M/\delta)}{2K} \right) = 2\frac{\delta}{2M} = \frac{\delta}{M} \tag{15}$$

since each $\ell(\hat{p}_n^k, x_k, y_k)$ is an independent variable, bounded in [0,1] with expectation $L(\mathcal{A}_\phi, n)$.

Now when sample complexity is less than $M$, we use a union bound to translate this to a high probability bound on error of $\hat{m}$, so that with probability at least $1 - \delta$:

$$|\hat{m} - m(\phi, \mathcal{D}, \varepsilon, \mathcal{A})| = \left| \sum_{n=1}^{M} [\hat{L}_n - \varepsilon]_+ - [L(\mathcal{A}_\phi, n) - \varepsilon]_+ \right| \tag{16}$$

$$\leq \sum_{n=1}^{M} \left| [\hat{L}_n - \varepsilon]_+ - [L(\mathcal{A}_\phi, n) - \varepsilon]_+ \right| \tag{17}$$

$$\leq \sum_{n=1}^{M} \left| \hat{L}_n - L(\mathcal{A}_\phi, n) \right| \tag{18}$$

$$\leq M \sqrt{\frac{\log(2M/\delta)}{2K}} \tag{19}$$

This gives us the first part of the claim.

We want to know that when the algorithm returns `tight`, the estimate can be trusted (i.e. that we set $M$ large enough). Under the assumption of large enough $K$, and by an application of Hoeffding, we have that

$$P\left( L(\mathcal{A}_\phi, M) - \hat{L}_M > \varepsilon/2 \right) \leq \exp\left( -2K\varepsilon^2 \right) \leq \exp\left( -2\frac{\log(1/\delta)}{2\varepsilon^2}\varepsilon^2 \right) = \delta \tag{20}$$

If $\hat{L}_M \leq \varepsilon/2$, this means that $L(\mathcal{A}_\phi, M) \leq \varepsilon$ with probability at least $1 - \delta$. By the assumption of decreasing loss, this means the sample complexity is less than $M$, so the bound on the error of $\hat{m}$ holds. □

## APPENDIX B  ALGORITHMIC DETAILS FOR ESTIMATING SAMPLE COMPLEXITY

Recall that $\varepsilon$ sample complexity ($\varepsilon$SC) is defined as

$$m_{\varepsilon\text{SC}}(\phi, \mathcal{D}, \mathcal{A}, \varepsilon) = \min\left\{ n \in \mathbb{N} : L(\mathcal{A}_\phi, n) \leq \varepsilon \right\}. \tag{21}$$

We estimate $m_{\varepsilon\text{SC}}$ via recursive grid search. To be more precise, we first define a search interval $[1, N]$, where $N$ is a large enough number such that $L(\mathcal{A}_\phi, N) \ll \varepsilon$. Then, we partition the search interval in to 10 sub-intervals and estimate risk of hypothesis learned from $D^n \sim \mathcal{D}^n$ with high confidence for each sub-interval. We then find the leftmost sub-interval that potentially contains $m_{\varepsilon\text{SC}}$ and proceed recursively. This procedure is formalized in Algorithm 2 and its guarantee is given by Theorem 5.

**Theorem 5.** *Let the loss function $L$ be bounded in $[0, 1]$ and assume that it is decreasing in $n$. Then, Algorithm 2 returns an estimate $\hat{m}$ that satisfies $m_{\varepsilon\text{SC}}(\phi, \mathcal{D}, \mathcal{A}, \varepsilon) \leq \hat{m}$ with probability at least $1 - \delta$.*

*Proof.* By Hoeffding, the probability that $|\hat{L}_n - L(\mathcal{A}_\phi, n)| \geq \varepsilon/2$, where $\hat{L}$ is computed with $S = 2\log(20k/\delta)/\varepsilon^2$ independent draws of $D^n \sim \mathcal{D}^n$ and $(x, y) \sim \mathcal{D}$, is less than $\delta/(10k)$. The algorithm terminates after evaluating $\hat{L}$ on at most $10k$ different $n$'s. By a union bound, the probability that $|\hat{L}_n - L(\mathcal{A}_\phi, n)| \leq \varepsilon/2$ for all $n$ used by the algorithm is at least $1 - \delta$. Hence, $\hat{L}_n \leq \varepsilon/2$ implies $L(\mathcal{A}_\phi, n) \leq \varepsilon$ with probability at least $1 - \delta$. □

## APPENDIX C  EXPERIMENTAL DETAILS

In each experiment we first estimate the loss-data curve using a fixed number of dataset sizes $n$ and multiple random seeds, then compute each measure from that curve. Reported values of SDL correspond to the estimated area between the loss-data curve and the line $y = \varepsilon$ using Riemann sums with the values taken from the left edge of the interval. This is the same as the chunking procedure of Voita & Titov (2020) and is equivalent to the code length of transmitting each chunk of data using a

---

**Algorithm 2:** Estimate sample complexity via recursive grid search

---

**Input:** Search upper limit $N$, parameters $\varepsilon$, confidence parameter $\delta$, data distribution $\mathcal{D}$, and learning algorithm $\mathcal{A}$.

**Output:** Estimate $\hat{m}$ such that $m_{\varepsilon SC}(\phi, \mathcal{D}, \mathcal{A}, \varepsilon) \leq \hat{m}$ with probability $1 - \delta$.

---

let $S = 2 \log(20k/\delta)/\varepsilon^2$, and let $[\ell, u]$ be the search interval initialized at $\ell = 1, u = N$.

**for** $r = 1$ **to** $k$ **do**
    Partition $[\ell, u]$ into 10 equispaced bins and let $\Delta$ be the length of each bin.
    **for** $j = 1$ **to** 10 **do**
        Set $n = \ell + j\Delta$.
        Compute $\hat{L}_n = \frac{1}{S} \sum_{i=1}^{S} \ell(\mathcal{A}(D_i^n), x_i, y_i)$ for $S$ independent draws of $D^n$ and test sample $(x, y)$.
        **if** $\hat{L}_n \leq \varepsilon/2$ **then**
            Set $u = n$ and $\ell = n - \Delta$.
            **break**

**return** $\hat{m} = u$, which satisfies $m_{\varepsilon SC}(\phi, \mathcal{D}, \mathcal{A}, \varepsilon) \leq \hat{m}$ with probability $1 - \delta$, where the randomness is over independent draws of $D^n$ and test samples $(x, y)$.

---

fixed model and switching models between intervals. Reported values of $\varepsilon SC$ correspond to the first measured $n$ at which the loss is less than $\varepsilon$.

All of the experiments were performed on a single server with 4 NVidia Titan X GPUs, and on this hardware no experiment took longer than an hour. All of the code for our experiments, as well as that used to generate our plots and tables, is included in the supplement.

## C.1 MNIST EXPERIMENTS

For our experiments on MNIST, we implement a highly-performant vectorized library in JAX to construct loss-data curves. With this implementation it takes about one minute to estimate the loss-data curve with one sample at each of 20 settings of $n$. We approximate the loss-data curves at 20 settings of $n$ log-uniformly spaced on the interval $[10, 50000]$ and evaluate loss on the test set to approximate the population loss. At each dataset size $n$ we perform the same number of updates to the model; we experimented with early stopping for smaller $n$ but found that it made no difference on this dataset. In order to obtain lower-variance estimates of the expected risk at each $n$, we run 8 random seeds for each representation at each dataset size, where each random seed corresponds to a random initialization of the probe network and a random subsample of the evaluation dataset.

Probes consist of two-hidden-layer MLPs with hidden dimension 512 and ReLU activations. All probes and representations are trained with the Adam optimizer (Kingma & Ba, 2015) with learning rate $10^{-4}$.

Each representation is normalized to have zero mean and unit variance before probing to ensure that differences in scaling and centering do not disrupt learning. The representations of the data we evaluate are implemented as follows.

**Raw pixels.** The raw MNIST pixels are provided by the Pytorch `datasets` library (Paszke et al., 2019). It has dimension $28 \times 28 = 784$.

**CIFAR.** The CIFAR representation is given by the last hidden layer of a convolutional neural network trained on the CIFAR-10 dataset. This representation has dimension 784 to match the size of the raw pixels. The network architecture is as follows:

```
nn.Conv2d(1, 32, 3, 1),
nn.ReLU(),
nn.MaxPool2d(2),
nn.Conv2d(32, 64, 3, 1),
nn.ReLU(),
nn.MaxPool2d(2),
```

```
        nn.Flatten(),
        nn.Linear(1600, 784)
        nn.ReLU()
        nn.Linear(784, 10)
        nn.LogSoftmax()
```

**VAE.**   The VAE (variational autoencoder; Kingma & Welling (2014); Rezende et al. (2014)) representation is given by a variational autoencoder trained to generate the MNIST digits. This VAE's latent variable has dimension 8. We use the mean output of the encoder as the representation of the data. The network architecture is as follows:

```
self.encoder_layers = nn.Sequential(
    nn.Linear(784, 400),
    nn.ReLU(),
    nn.Linear(400, 400),
    nn.ReLU(),
    nn.Linear(400, 400),
    nn.ReLU(),
)
self.mean = nn.Linear(400, 8)
self.variance = nn.Linear(400, 8)

self.decoder_layers = nn.Sequential(
    nn.Linear(8, 400),
    nn.ReLU(),
    nn.Linear(400, 400),
    nn.ReLU(),
    nn.Linear(400, 784),
)
```

## C.2   PART OF SPEECH EXPERIMENTS

We follow the methodology and use the official code[2] of Voita & Titov (2020) for our part of speech experiments using ELMo (Peters et al., 2018) pretrained representations. In order to obtain lower-variance estimates of the expected risk at each $n$, we run 4 random seeds for each representation at each dataset size, where each random seed corresponds to a random initialization of the probe network and a random subsample of the evaluation dataset. We approximate the loss-data curves at 10 settings of $n$ log-uniformly spaced on the range of the available data $n \in [10, 10^6]$. To more precisely estimate $\varepsilon$SC, we perform one recursive grid search step: we space 10 settings over the range which in the first round saw $L(\mathcal{A}_\phi, n)$ transition from above to below $\varepsilon$.

Probes consist of the MLP-2 model of Hewitt & Liang (2019); Voita & Titov (2020) and all training parameters are the same as in those works.

---

[2]https://github.com/lena-voita/description-length-probing

