# OpenReview forum: "Evaluating representations by the complexity of learning low-loss predictors"
_ICLR.cc/2021/Conference — Reject_

### Official Review · AnonReviewer1 · 2020-10-28
**Good discussion and motivation, well grounded methods, accept**

**Rating:** 7
**Confidence:** 4

**Review:**

The submission addresses the problem of representation evaluation from the perspective of efficient learning of downstream predictors. Leveraging the introduced loss-data curve framework, the paper studies and demonstrates the limitations of the existing methods in terms of their implicit dependency on evaluation dataset size. Motivated by practicality and interpretability of the measures for choosing the best representations, the paper introduces two novel methods, $\epsilon$ sample complexity ($\epsilon$SC) and surplus description length (SDL), which are well-motivated and supported both theoretically and empirically. The paper also delivers efficient implementation.

The paper has excellent motivation and discussion about how existing methods are inconsistent with representation quality, computational complexity and are not robust when evaluation dataset size changes—thus deeming them inapt to answer practical questions. For example, how hard is it to learn a predictor on the given representation? How many samples with given the representation are needed to achieve a specified performance? Is it even possible to reach a specified performance given this representation?
The proposed methods are designed to handle these questions and are robust to the evaluation dataset size secured against precipitate decisions on which representation is best. The paper considers the best representation to be the one that allows the most efficient learning of a downstream predictor.

Although I do not immediately see weak points, yet the following things should be taken into account when applying the proposed methods — the assumption of monotonically improving predictors, the dependency of the proposed and existing methods for representation evaluation on the probe design, and potential disagreement between $\epsilon$SC and SDL (as seen in Figure 3 for $\epsilon=0.5$).
Although very plausible, the assumption might not hold when there is a sufficient shift between training and validation data. There might be situations when representations may overfit to training data and not reflect properties of validation set points, potentially causing problems in both proposed methods. While dependency on the predictors is addressed in the paper as a separate problem and a direction for future work, it would help if the authors discussed the cases when there is a disagreement between $\epsilon$SC and SDL.

Overall, this submission seems to be a solid contribution and should be accepted as it rigorously addresses a problem of representation evaluation, which is of broad interest to the ICLR community.

---

> ### Author Response · Authors · 2020-11-12
> **Author response**
>
> We thank the reviewer for a thorough and insightful review. We especially appreciate that the reviewer recognizes the importance of the topic and the practicality of our proposed measures. We address some questions below.
>
> **“Although very plausible, the assumption might not hold...”**
>
> We agree that the assumption of monotonically improving predictors is important for practical implementations! However, we wish to emphasize that this monotonic improvement is with respect to additional training data, not update steps; essentially, an algorithm trained on more data should make better (or at least, not worse) predictions in expectation. This property holds for most actual algorithms.
>
> If there is a domain shift between the training and validation distributions, however, this assumption may become a problem. We believe this is an important problem but rather outside the scope of this work. We leave the study of the interaction of representation and out-of-domain generalization to other works, for example [3].
>
> [3]: Arjovsky, M., Bottou, L., Gulrajani, I., & Lopez-Paz, D. (2019). Invariant Risk Minimization. ArXiv, abs/1907.02893.
>
>
> **“it would help if the authors discussed the cases when there is a disagreement between ϵSC and SDL”**
>
> Thank you for pointing this out, as it's certainly worth discussing. SDL and εSC can disagree in their evaluations because they fundamentally measure different things. SDL measures the amount of information in bits that an algorithm has extracted from the data, whereas εSC measures this quantity in samples. From one perspective, εSC is simply a coarse approximation to SDL, and in fact for the research setting we encourage the use of SDL.
>
> However in practice, which measure is most appropriate depends on the application. SDL measures the additional cost of training a model online instead of ahead of time, and might better reflect the cost structure of continuous training of a deployed model. On the other hand, εSC measures the number of samples until a model performs well, and might be the correct measure for a company which has to pay for each human annotation of a data point.

---

### Official Review · AnonReviewer4 · 2020-10-30

**Rating:** 4
**Confidence:** 3

**Review:**

*Summary of the paper:
The paper discusses new measures to evaluating the quality of representations.
They present two new measures namely SDL and $\epsilon$-sample complexity, describe their benefits and apply experimental study, comparing these measures to standard baselines.

* Comments
The suggested measures are simple modifications of existing baselines (MDL,MI).
The authors emphasize that their measures are independent of dataset size $n$, but this only applies when $n$ is large enough and when $\epsilon$ is a good estimate of the conditional entropy.

The authors also mention two issues regarding existing measures (MDL,MI)
- They are insensitive to statistical complexity
- They are insensitive to computational complexity
while these are true for the theoretical measures, I don't see what is the benefit of the current measures in this context. This is since old and new measures are leaning on an approximate  computation of $L(A,i)$.


*Summary of review
I don't see any clear benefit of the suggested measures.

---

> ### Author Response · Authors · 2020-11-12
> **Author response**
>
> **“only applies when n is large enough...”**
>
> In fact the measures are always independent of dataset size; however, if ε is set too low or if little data is available, it may only be possible to compute a lower bound. We consider it a feature of our measures that they are able to report when there is not enough data to achieve the target loss or rank the representations. In a practical setting, a measure which returns an answer when it does not have sufficient data to do so is a serious liability. Note that for our measures to be defined, ε can be any value larger than the conditional entropy and does not need to be an estimate of it.
>
> If the loss target ε is set too low, it may take arbitrarily much data for an algorithm to achieve it, or it may be unreachable. This indicates that the representation being evaluated is of too poor a quality to enable learning an ε-loss predictor. If ε was set according to our recommendations in Section 4.3 paragraph 2, this indicates that learning an ε-loss predictor is actually harder using this representation than it is using the raw data. We believe this would be sufficient reason to rule out the representation in question.
>
>
> **“old and new measures are leaning on an approximate computation of L(A,i)”**
>
> First, we think the reviewer may have misread Section 3.2 — these critiques apply only to mutual information, not to MDL.
>
> Second, we understand that MDL, MI and our measures use an empirical approximation of L(A,i). But this in no way makes them all the same. As we explain in Section 3.1, VA and MDL suffer from sensitivity to the size of the evaluation dataset which our methods do not and which is not related to the shared use of the approximation of L(A,i). And as we describe in Section 3.2, mutual information cannot be computed with a finite sample and model. This means that substituting L(A, n) for mutual information is incorrect even for the true value of L(A, n). Finally, as we describe in Section 3.3 all prior methods lack an a priori notion of success which makes them liable to rank representations without sufficient data to do so properly.
>
>
> **“I don't see any clear benefit of the suggested measures.”**
>
> We hope that the above response clarifies the benefits of our approach over prior methods for the reviewer. If not, we are happy to answer any more specific questions they might have about the benefits of our proposed methods.

---

### Official Review · AnonReviewer3 · 2020-11-01
**Review for "Evaluating representations by the complexity of learning low-loss predictors"**

**Rating:** 4
**Confidence:** 3

**Review:**

The authors address the issue of evaluating the quality of representations based on performance of a classifier for a downstream task. The premise is that conventional metrics for the downstream classifier such as Validation Accuracy, MDL or MI are flawed due to dependence on the size of the dataset (VA, MDL) or because they ignore statistical/computational complexity of the classifier (MI).
The alternatives proposed in the paper SDL/\epsilon-SC, which attempt to measure the complexity of learning the classifier to \epsilon-tolerance.

The core premise is reasonable. The metric/evaluation of the representation should not depend on the amount of data available for the downstream task. However, it seems like the proposed alternative replaces an arbitrary dataset size, with an arbitrary loss threshold (small values of \epsilon may lead no measurement (infinity for both SDL and \epsilon-SC), while larger values of \epsilon may also lead to arbitrariness. The authors do say that the they want an evaluation measure which depends on the data distribution, but seem to not pay sufficient attention to the "distribution" part (measurements are noisy, and we should understand what is noise, and what is a significant improvement, possibly by computing confidence intervals). The authors specifically mention a downside of MI is that it ignores computational complexity of the classifiers, but that seems to be a valid complaint of their approach as well (presumably more (computationally) complex classifiers can achieve a loss threshold), though their approach does address the issue with statistical complexity.
There are other issues - there are typically many tasks that we might want to use a common representation for. The arbitrariness of epsilon makes combining evaluations across different tasks difficult. Further, we typically have a fixed amount of data for each downstream task - it would seem that the approach proposed here would require an arbitrary amount of data for each task, which may not be feasible.

---

> ### Author Response · Authors · 2020-11-12
> **Author response**
>
> We thank the reviewer for their feedback. We are glad to note that they appreciate our critiques of existing methods. We address specific points from their review below.
>
> **"replaces an arbitrary dataset size, with an arbitrary loss threshold"**
>
> Our paper argues that one common goal in representation learning is to achieve good performance with the least data possible. While this is not the goal for every project, it is an important use case that to date lacked good tools for analysis. Our goal is to add a tool to the toolbox of the practitioner.
>
> We also believe that it is in general easier to know what success looks like (for example, achieving a 5% error rate) than to know how much data it will take to achieve meaningfully good performance. We provide guidance for selecting ε in Section 4.3.
>
>
> **“we should understand what is noise, and what is a significant improvement, possibly by computing confidence intervals”**
>
> We completely agree that noisy measurements are a concern in representation evaluation! Our exposition in the paper uses expected loss throughout, which in our experiments we approximate using 8 bootstrap-sampled datasets and probe initializations to construct every point in Figure 2, and 4 samples per point in Figure 3. Our open-source library similarly computes expected values using many samples. We can add confidence intervals to the plots and tables.
>
>
> **"a downside of MI is that it ignores computational complexity of the classifiers, but that seems to be a valid complaint of their approach as well"**
>
> The issues with the way that MI ignores computational complexity is that the measure is (1) not computable, and (2) has no practical relevance to training a useful model. By contrast, our measures embrace computational limitations: instead of attempting (and failing) to measure  the best performance possible on the task, they measure the performance realizable by a practical algorithm as instantiated by the probe. This is related to V-information [1] or the decodable information bottleneck [2].
>
> [1]: Xu, Y., Zhao, S., Song, J., Stewart, R., & Ermon, S. (2020). A Theory of Usable Information Under Computational Constraints. ArXiv, abs/2002.10689.
> [2]: Dubois, Y., Kiela, D., Schwab, D.J., & Vedantam, R. (2020). Learning Optimal Representations with the Decodable Information Bottleneck. ArXiv, abs/2009.12789.
>
>
> **“epsilon makes combining evaluations across different tasks difficult”**
>
> In general, combining evaluations across different tasks is a somewhat subjective enterprise, as losses for different tasks will typically be on different scales. To our understanding, there is not a generally-agreed-upon approach for solving this problem.
>
> For our methods, we recommend following the protocol described in Section 4.3 paragraph 2: for each task, train a large model on the raw representation of the dataset and use its final loss as ε. This gives a baseline of performance on the raw representation. Using SDL gives a measure of the additional number of bits incurred by transmitting data according to a learning algorithm on representation φ rather than according to this fully-trained model.
>
> These measures could be combined into a composite measure of performance in various ways similar to any other score. For example, the SDL measures can be added together, giving a score in terms of "surplus bits relative to raw baseline", or each representation could be ranked by how often it outperforms every other.
>
>
> **“it would seem that the approach proposed here would require an arbitrary amount of data”**
>
> If the loss target ε is set too low, it may indeed take arbitrarily much data for an algorithm to achieve it, or it may be unreachable. This indicates that the representation being evaluated is of too poor a quality to enable learning an ε-loss predictor. If ε was set according to Section 4.3, this indicates that learning an ε-loss predictor is actually harder using this representation than it is using the raw data. We believe this would be sufficient reason to rule out the representation in question.
>
> Existing approaches such as MDL effectively assume ε=0, requiring even more data to create a consistent measurement.

---

### Decision · Program_Chairs · 2021-01-07
**Final Decision**

**Decision:**

Reject

**Comment:**

The paper studies the problem of evaluating representations and proposes two new metrics: surplus description length and epsilon sample complexity.

Pros:
- A good overview of existing methods and their corresponding weaknesses (i.e. sensitivity to dataset size and insensitivity to representation quality and computational complexity).
- The proposed procedures seem to be well-supported conceptually.
- Has an efficient implementation.

Cons:
- While the theoretical results in the Appendix are appreciated and do provide some insight into the procedures, they more or less seem straightforward and don't answer some important questions (i.e. what is the sample size necessary in terms of epsilon? is it exponential in some dimension?).
- More insight could have been provided into where the noisy measurements come from in these metrics as there appear to be many components in the calculations that could be contributing to the noise (i.e. dataset distribution, dataset size, bootstrap samples, probe initializations, etc).
- The methods make an assumption that the performance is monotonic in the dataset size, which is often not the case (i.e. there is a subfield regarding removing noisy label examples to improve performance; moreover there are investigations in the active learning literature that suggest sometimes performance degrades with more training data).
- It appears that the proposed metrics are based on data efficiency (i.e. least number of samples to get obtain a desired performance). However, such may have more of a dependence on the distribution of the data and how the examples are chosen (i.e. they can be actively chosen) moreso than the actual representation. This may or may not be an issue but may deserve at least some discussion.

Overall, the reviewers appreciated the new methods proposed and how they relate and improve upon previous methods; however, as currently presented, most were unconvinced about its significance which was a key reason for rejection.